# Pegylated Liposomal Alendronate Biodistribution, Immune Modulation, and Tumor Growth Inhibition in a Murine Melanoma Model

**DOI:** 10.3390/biom13091309

**Published:** 2023-08-26

**Authors:** Md. Rakibul Islam, Jalpa Patel, Patricia Ines Back, Hilary Shmeeda, Raja Reddy Kallem, Claire Shudde, Maciej Markiewski, William C. Putnam, Alberto A. Gabizon, Ninh M. La-Beck

**Affiliations:** 1Department of Immunotherapeutics and Biotechnology, Jerry H. Hodge School of Pharmacy, Texas Tech University Health Sciences Center, Abilene, TX 79601, USA; rakibul.islam@ttuhsc.edu (M.R.I.); jalpa.patel@ttuhsc.edu (J.P.); pback@ttuhsc.edu (P.I.B.); ces14a@acu.edu (C.S.); maciej.markiewski@ttuhsc.edu (M.M.); 2Nano-Oncology Research Center, Oncology Institute, Shaare Zedek Medical Center, Jerusalem 9103102, Israel; hilary@szmc.org.il; 3Department of Pharmacy Practice, Jerry H. Hodge School of Pharmacy, Texas Tech University Health Sciences Center, Abilene, TX 79601, USA; rajareddy.kallem@ttuhsc.edu (R.R.K.); trey.putnam@ttuhsc.edu (W.C.P.); 4Clinical Pharmacology and Experimental Therapeutics Center, Jerry H. Hodge School of Pharmacy, Texas Tech University Health Sciences Center, Dallas, TX 75235, USA; 5Department of Pharmaceutical Science, Jerry H. Hodge School of Pharmacy, Texas Tech University Health Sciences Center, Dallas, TX 75235, USA; 6Faculty of Medicine, The Hebrew University of Jerusalem, Jerusalem 9112102, Israel

**Keywords:** PD-1, immune checkpoint, liposomal alendronate, immunotherapy, cancer, tumor, melanoma

## Abstract

While tumor-associated macrophages (TAM) have pro-tumoral activity, the ablation of macrophages in cancer may be undesirable since they also have anti-tumoral functions, including T cell priming and activation against tumor antigens. Alendronate is a potent amino-bisphosphonate that modulates the function of macrophages in vitro, with potential as an immunotherapy if its low systemic bioavailability can be addressed. We repurposed alendronate in a non-leaky and long-circulating liposomal carrier similar to that of the clinically approved pegylated liposomal doxorubicin to facilitate rapid clinical translation. Here, we tested liposomal alendronate (PLA) as an immunotherapeutic agent for cancer in comparison with a standard of care immunotherapy, a PD-1 immune checkpoint inhibitor. We showed that the PLA induced bone marrow-derived murine non-activated macrophages and M2-macrophages to polarize towards an M1-functionality, as evidenced by gene expression, cytokine secretion, and lipidomic profiles. Free alendronate had negligible effects, indicating that liposome encapsulation is necessary for the modulation of macrophage activity. In vivo, the PLA showed significant accumulation in tumor and tumor-draining lymph nodes, sites of tumor immunosuppression that are targets of immunotherapy. The PLA remodeled the tumor microenvironment towards a less immunosuppressive milieu, as indicated by a decrease in TAM and helper T cells, and inhibited the growth of established tumors in the B16-OVA melanoma model. The improved bioavailability and the beneficial effects of PLA on macrophages suggest its potential application as immunotherapy that could synergize with T-cell-targeted therapies and chemotherapies to induce immunogenic cell death. PLA warrants further clinical development, and these clinical trials should incorporate tumor and blood biomarkers or immunophenotyping studies to verify the anti-immunosuppressive effect of PLA in humans.

## 1. Introduction

Bisphosphonates have been commercially available since 1977 for the treatment of various bone diseases, such as osteoporosis, Paget’s disease, hypercalcemia, and cancer bone lesions. Bisphosphonates are structurally similar to pyrophosphates, containing phosphate groups that enable the drug to bind to the bone mineral hydroxyapatite, where the drug is delivered to osteoclasts through their bone resorptive activity [1,2]. The first-generation bisphosphonates (e.g., clodronate) act by forming a toxic ATP analog that inhibits ATP-dependent enzymes, leading to the initiation of apoptosis and cell death [3]. The second- and third-generation bisphosphonates (e.g., alendronate, zoledronate) have side chains containing nitrogen and are classified as amino-bisphosphonates (N-BPs). Unlike the first-generation bisphosphonates, N-BPs act by inhibiting farnesyl pyrophosphate synthase, an enzyme in the mevalonate pathway critical for the production of cholesterol and sterols [4]. Evidence from clinical studies evaluating N-BPs for the treatment or prevention of metastatic bone disease in cancer patients suggests that they may have antitumor effects that are not related to their skeletal effects [5,6,7]. In the Adjuvant Zoledronate to Reduce Recurrence in Early Breast Cancer (AZURE) trial, post-menopausal breast cancer patients treated in the adjuvant setting with a combination of zoledronate and standard-of-care chemotherapy experienced a significant improvement in disease-free survival compared to chemotherapy alone [8,9].

A clear limitation of all bisphosphonates in free form, with or without an amino group in side chains, is their rapid clearance by the kidneys and sequestration in the bone, which severely limits systemic exposure and applications for the treatment of cancer. The encapsulation of bisphosphonates in liposomes to overcome the unfavorable pharmacokinetics and target clodronate to block the reticuloendothelial system (Kupffer cells and other tissue-fixed macrophages) was first reported by Van Rooijen and coworkers with a multilamellar liposomal formulation of clodronate [10]. Since then, clodronate liposomes have been found to deplete systemic macrophages in a variety of murine models of human diseases, including cancer, where non-specific macrophage ablation induced by clodronate liposomes was found to have anti-tumoral effects [11]. Although the ablation of tumor-associated macrophages (TAMs) may be therapeutic due to their ability to promote tumor cell proliferation, neoangiogenesis, and invasion, macrophage ablation may not be desirable since they also phagocytose cancer cells, prime and activate T cells against tumor antigens, and remodel the tumor immunologic milieu. In addition, blockade of Kupffer cells decreases their bacterial clearance activity and may pose a significant risk in the control of infectious diseases [12,13]. The pro-tumoral and anti-tumoral functions of macrophages are associated with their polarization state: M2 macrophages promote tumor growth, while M1 macrophages inhibit tumor growth, and the repolarization of macrophages from the M2 to the M1 phenotype leads to a tumor-suppressive effect [14,15]. Delivery of the more potent aminobisphosphonates encapsulated in stealth liposomes, and in much smaller doses than clodronate, has drastically changed the PK and PD and significantly reduced their systemic off-target toxicity towards Kupffer cells and other non-TAMs [16].

Alendronate is one of the most potent N-BPs [17]; it has direct tumoricidal activity and modulates the function of macrophages and other myeloid cells in vitro [18,19,20], suggesting its potential as an immunotherapeutic agent. Our aim was to repurpose alendronate in liposomal form as a broad-spectrum immunotherapeutic agent for cancer. Similar immunotherapy strategies using liposomal drug delivery to modulate the tumor immunologic milieu have been published for other cancer models [21], further supporting the feasibility of our approach. We selected a long-circulating liposomal carrier formulation that confers “stealth” properties to increase the tumor tissue accumulation of the drug payload via the enhanced permeability and retention effect [22]. To prevent leakage of the drug in circulation, we used high phase-transition phospholipids and cholesterol as the main liposome components. To achieve long circulation, we decreased the vesicle size to a diameter of ~100 nm and added a small amount (5%) of a pegylated phospholipid (polyethylene glycol-1,2-distearoyl-sn-glycero-3-phosphoethanolamine; PEG-DSPE). The liposome composition was similar to that of the clinically approved and widely used formulation of pegylated liposomal doxorubicin (Doxil^®^), which facilitates rapid clinical translation. Passive tumor targeting of these long-circulating liposomes has been documented in humans [23], although its extent is highly variable [24]. Here, we examined the immune modulatory effects of this pegylated liposomal alendronate (PLA) formulation on polarized and unpolarized bone marrow-derived macrophages, and its effects on remodeling of the tumor immunologic milieu and antitumor efficacy in comparison with a standard of care immunotherapy, a PD-1 immune checkpoint inhibitor, in a syngeneic mouse model of melanoma.

## 2. Materials and Methods

### 2.1. Key Chemicals and Formulations

Hydrogenated soy phosphatidylcholine (HSPC) and methoxy-polyethylene glycol (PEG_2000_)-distearoyl-phosphoethanolamine (mPEG_2000_-DSPE) were from Lipoid (Ludwigshafen, Germany), cholesterol was from Sigma (St. Louis, MO, USA), alendronate was from Teva (Tel Aviv, Israel), and indocarbocyanine dye 1,1’-dioctadecyl-3,3,3’,3’-tetramethylindocarbocyanine perchlorate (DiI) was from Invitrogen (Darmstadt, Germany). Purified in vivo grade anti-PD1 (clone RMP1-14) and rat IgG isotype control antibodies were purchased from Bio X Cell (Lebanon, NH, USA). Heat-inactivated fetal bovine serum (HI-FBS) (MT35016CV), non-essential amino acid (25-025CI), L-glutamine (25005CI), and sodium pyruvate (25000CI) were from Corning^®^ (Corning, NY). HEPES (15630-080) was from GIBCO, 5% dextrose injection USP (S5104-5384) was from B.BRAUN, and rmGM-CSF (200-15) and IL-4 (200-18) were from Shenandoah.

Liposomes were synthesized with an approximate molar composition of 55% hydrogenated soy phosphatidylcholine (HSPC), 40% cholesterol, and 5% methoxy-polyethylene glycol (mPEG_2000_)-distearoyl-phosphoethanolamine (DSPE) [19], using the standard ethanol injection technique and mixing them into an aqueous buffer, followed by extrusion. To encapsulate alendronate, lipids were dissolved in ethanol and injected into an aqueous buffer containing 250 mM of ammonium alendronate, pH 6.5–7.0, at a 1:4 *v*/*v* ratio. The resulting multilamellar liposome suspension was extruded at 60 °C through polycarbonate membranes of a 0.08–0.10 µm pore size to obtain unilamellar liposomes. Non-encapsulated alendronate was removed by dialysis [25]. The liposomes were then sterilized by filtration through 0.22 μm filters and stored in vacuum-sealed, sterile, 15 mL glass tubes at 4 °C. The final concentrations of the phospholipids and alendronate were quantified by phosphorus analyses with the Bartlett method [26] after both phosphate-containing components were separated by the Bligh and Dyer or Folch partition method [27] (phosphates from phospholipids in organic phase and phosphates from alendronate in water phase). The accuracy of this method was previously verified at the National Cancer Institute’s Nanotechnology Characterization Laboratory by quantitation of alendronate with an HPLC-based method [28]. For biodistribution studies, DiI was post-loaded into the PLA formulation as follows: the DiI was prepared as a 2.5 mg/mL stock solution in ethanol and added to liposomes at a 0.4% molar ratio of the phospholipids, with shaking for 30 min at 50 °C, then cooled and centrifuged at 4 °C, 3000 rpm for 15 min. The DiI-PLA was re-sterilized, and the sample was passed through a Sepharose 6B column (GE Healthcare) to test for free DiI. Dil-labelled liposomes were stable for 6 months at 4 °C. All formulations were endotoxin-free by the LAL assay (Hay Labs, Rehovot, Israel). Placebo liposomes were a gift from Lipomedix Pharmaceuticals (Jerusalem, Israel).

### 2.2. Cell Culture

B16-OVA melanoma cells, originally developed from B16-F10 parental cells [29], were a gift from Dr. Laurence Wood. The cells were cultured in Dulbecco’s Modified Eagle Medium (DMEM) supplemented with 10% heat-inactivated fetal bovine serum (HI-FBS) and 1% penicillin/streptomycin. Aliquots of the cells used in the experiments were tested for mycoplasma and confirmed to be negative.

Bone marrow-derived macrophages (BMDMs) were differentiated from the femur and tibia of C57BL/6 male mice using DMEM 10-17-CV supplemented with 20% premium heat-inactivated FBS, 30% L929 cell supernatant, 1% L-glutamine, 1% sodium pyruvate, and 1% penicillin/streptomycin. The cells were incubated at 37 °C/5% CO_2_ for 8 days, with half of the media replaced on day 4. The BMDMs were re-plated for polarization using IMDM media supplemented with 10% premium heat-inactivated FBS and 1% penicillin/streptomycin. Polarization was performed using 100 ng/mL lipopolysaccharide (LPS) for M1 macrophages and 20 ng/mL IL-4 for M2 macrophages. The cells were incubated with polarization stimuli for 12 h before treatment with PLA, and free alendronate, placebo liposome, and dextrose were added. The liposome concentrations were based on the Doxil C_max_ phospholipid concentration reported in patients (55.7 µM) [30], corresponding to 19.6 µM of alendronate.

### 2.3. In Vitro Cytotoxicity

B16-OVA melanoma cells and bone marrow-derived macrophages were cultured as above. All cells were then seeded at a density of 15,000 cells per 50 µL media in 96-well plates. After a 24 h incubation period, the cells were treated with different concentrations of doxorubicin (positive control), free alendronate, PLA, or anti-PD1 for 48 h. The highest concentrations used for each treatment were doxorubicin 800 nM, alendronate 800 nM, anti-PD1 50 µg/mL, and PLA 5600 nM. After a further 24 h incubation period, the cytotoxicity was assessed using MTT assay kits (#G4000, Promega and #4890050K, R&D Systems), and absorbance at 570 nm was measured using the BioTek Synergy HT and analyzed with Gen5 software.

### 2.4. RT-qPCR

Polarized and unpolarized BMDMs were derived and treated with PLA or controls for 24 h, and then the cells were collected with RLT plus buffer containing β-mercaptoethanol for RT-qPCR (*n* = 4). The RNA was extracted using the RNeasy^®^ plus mini kit (QIAGEN, Hilden, Germany), and the cDNA was produced using the High-Capacity RNA-to-cDNA kit (Applied Biosystems, Beverly Hills, CA, USA). Real-time quantitative PCR was performed using PowerUp Sybr Green Master mix, primer mixes (8 µM), and StepOnePlus^TM^ (Applied Biosystems) following 40 amplification cycles. We assessed the M1 and M2 macrophage markers: Arg-1 [31], iNOS [32], IL-6 [33], CXCL10 [34], IL-10 [35], and TGF-β [35]. The relative mRNA expression was normalized to Ubc [36] and calculated using the 2^∆∆Ct^ method and RT^2^ profiler PCR Array Data Analysis (QIAGEN). The primer sequences are in Appendix A.

### 2.5. Animals

Male and female 6–8-week-old C57BL/6 mice were purchased from Jackson Laboratory (Bar Harbor, ME, USA) and acclimatized for at least one week before initiating the in vivo studies. The animals were housed under standard conditions and cared for at the Texas Tech University Health Sciences Center (TTUHSC) animal care facility (Abilene, TX, USA), according to the Institutional Animal Care and Use Committee guidelines, and all procedures were conducted under an approved protocol.

For the biodistribution studies, the mice were subcutaneously implanted with 5 × 10^5^ B16-OVA cells on the rear flank. Tumor growth was monitored with a vernier caliper, and the tumor volume was estimated using the following equation: Volume = (AxB^2^)/2, where A = largest diameter, and B = smaller diameter. When the tumors reached 300 mm^3^, the mice were treated intravenously with either free alendronate or DiI-labeled pegylated liposomal alendronate (PLA) at 4 mg/kg alendronate equivalent, via tail vein injection. Animals were euthanized 1, 24, or 48 h later, for collection of their blood and tissues (tumors, tumor-draining inguinal lymph nodes, non-tumor-draining axillary lymph nodes, spleen, liver, lungs, and kidneys). The blood was processed to obtain the serum, and the tissue was embedded in an optimal cutting temperature (OCT) compound for fluorescence microscopy (only tissues) or snap-frozen. All samples were stored at −80 °C until analyses.

For the tumor growth studies, the mice were subcutaneously implanted with 1 × 10^6^ B16-OVA cells, the tumor size was monitored as described above, and treatment was initiated when the tumors were ~50 mm^3^. The mice were then randomized to one of three treatment groups: PLA at an alendronate dose of 4 mg/kg, anti-PD1 at a 10 mg/kg dose, or placebo controls (5% dextrose vehicle and IgG isotype). Each active treatment group also received the placebo control for the other treatment. PLA or vehicle was administered weekly by tail vein injection, followed by PD1-blockading monoclonal antibody (anti-PD1) or isotype control intraperitoneally every 48 h. The animals received two doses of PLA and three doses of anti-PD1. The animals were euthanized when humane endpoints were reached, and the tissues were collected and processed for bioanalytical assays, fluorescence microscopy, flow cytometry, and histopathology studies.

The animal weight was monitored as a measure of overall systemic toxicity. For organ-specific toxicity, tissues were collected at euthanasia and fixed with 10% formalin, then embedded in paraffin blocks; 5 µm sections were cut from each block and stained with hematoxylin and eosin (H&E) for histopathology. The slides were scanned and digitalized (Scan-Scope, Leica). Blinded histopathologic evaluation was performed by a pathologist (MM).

### 2.6. Alendronate Quantitation

Alendronate is a small polar and ionic compound that is not retained on most reversed phase columns; therefore, a derivatization method was employed using trimethylsilyldiazomethane to improve the chromatographic retention time and peak shape [37]. Alendronate concentrations in mouse tissue and serum were determined using a UHPLC-MS/MS analytical method that included the Shimadzu LC-30 AD Nexera Series UHPLC system (Shimadzu Corporation, Kyoto, Japan) and the Sciex QTRAP 5500 mass spectrometer (Foster City, CA, USA) equipped with the Turboionspray™ interface. Stock solutions of alendronate and alendronate-d_6_ internal standard (IS) and their secondary dilutions were prepared in water. The tissue samples were homogenized in four volumes of LC-MS-grade water using the PRO Scientific Inc. mechanical homogenizer (Oxford, CT, USA). The sample preparation involved a derivatization procedure where a 50 µL aliquot of tissue homogenate was placed into a 1.5 mL Eppendorf micro-centrifuge tube, and 5 µL of IS (2 µg/mL) solution was added and mixed, then diluted with 350 µL of LC-MS-grade water and 10 µL 1 M hydrochloric acid (HCl). The samples were loaded on to a Waters Oasis WAX 1 cc/30 mg solid phase extraction cartridge, which was pre-conditioned with 1 mL of methanol, followed by 1 mL of 20 mM HCl. Alendronate and IS were eluted into glass vials using 2 mL of trimethylsilyldiazomethane derivatizing reagent, followed by 1 mL of methanol. The derivatization reagent was prepared in the dark immediately before the derivatization procedure. The derivatization reagent was prepared by adding 9.75 mL of methanol/water (3:0.25; *v*/*v*) to 3 mL aliquot of trimethylsilyldiazomethane under nitrogen, followed by mixing and storage, under nitrogen and in the dark. The final alendronate and IS eluates were stored in the dark in glass vials for 30 min to complete the reaction. The reaction mixture was dried in a SpeedVac (Labconco, Kansas City, MO, USA) at −40 °C and reconstituted using 300 µL of water, acetonitrile, and formic acid (85:15:0.1, *v*/*v*/*v*). A 5 µL injection volume was employed.

Chromatographic separation was achieved using a Phenomenex Kinetex 2.6 µm, F5 100A, 2.1 × 150 mm analytical column, which was maintained at 40 ± 2 °C. Gradient elution was used where mobile phase A (MP-A) was 0.1% formic acid in water and mobile phase B (MP-B) was 0.1% formic acid in acetonitrile. A flow rate of 0.3 mL/min was employed. The gradient program started at 10% MP-B, which was maintained for the initial 2 min, then elevated to 75% MP-B at 3 min, which was held constant for the subsequent 2 min. The gradient program then changed back to 10% MP-B at 5.5 min for column re-equilibration over 7.5 min (total run time 7.5 min). Mass spectrometric detection of alendronate and IS was conducted using positive electrospray ionization and multiple-reaction monitoring. The precursor to product ion transition pairs were 348.2 to 163 *m*/*z* for alendronate, and *m*/*z* 354.2 to 168.1 for IS. Quadrupoles Q1 and Q3 were set to unit resolution. The standard curve was linear (r > 0.995) over the concentration range of 5–2000 ng/mL. Additionally, the calibration curve and quality control samples were analyzed with each run for accuracy throughout the analyses. The concentration of each of the calibration curve standards, when back-calculated, was determined to be within ± 15% from its nominal value, except at LLOQ, where it was determined to be within ± 20%. The back-calculated concentrations for the quality control samples were determined to be within ± 15% of their nominal value. Representative chromatograms are in Appendix A. The analytical data obtained were processed using Sciex’s Analyst software™ (version 1.6.2).

### 2.7. Fluorescence Microscopy

The tissues (liver, spleen, tumor, tumor-draining inguinal lymph node, non-tumor draining axillary lymph node, lung, and kidney) were embedded in OCT and frozen tissue blocks were sectioned at a 5 µm thickness. The tissue sections were mounted on glass slides and air-dried for 10 min, followed by incubation with DAPI for nuclear staining. Images were acquired for 8–10 randomly selected areas of each tissue (except the lymph nodes, for which 3 images captured the entire tissue area) at 20× using the confocal microscope (NIKON A1), and the DiI fluorescence was quantified using NIS Element software (https://www.microscope.healthcare.nikon.com/products/software/nis-elements, accessed on 1 August 2023).

### 2.8. Flow Cytometric Immunophenotyping

The tumors were excised, minced, and incubated for enzymatic digestion for 20 min, then neutralized with media containing FBS and passed through 40 μm strainers to obtain single-cell suspensions [38,39]. Spleens and lymph nodes were mechanically dissociated and passed through a 40 μm cell strainer. Red blood cells were lysed with ammonium–chloride–potassium (ACK) lysis solution. Single-cell suspensions were counted, and viability was assessed using trypan blue exclusion assay (Vi-Cell XR, Beckman Coulter Inc., Brea, CA, USA).

Three million cells from each sample were stained to enumerate and assess the functionality of the T cells (CD45, CD3, CD8a, and CD4) and myeloid cells (CD45, CD11b, CD11c, F4/80, MHC II, CD205, CD206, CD8a, and Gr1 or Ly6G, and Ly6C), similar to previously published methods [38,39]. The details of the gating strategies can be found in Appendix A. The samples were analyzed with an LSR Fortessa flow cytometer (BD Biosciences, San Jose, CA, USA), and the results were analyzed using FlowJo software (Tree Star Inc., Ashland, OR, USA, https://www.flowjo.com/, accessed on 1 August 2023). One million events were acquired, and dead cells were excluded via a fixable viability dye (BioLegend, cat. 423103). All flow cytometry antibodies were from BioLegend.

### 2.9. Statistical Analyses

The tumor volume and progression-free survival were evaluated as efficacy endpoints. The sample size was calculated to enable at least 80% power to detect a 50% difference in tumor size between the PLA versus the control placebo treatment groups. Two-way ANOVA was used to compare the tumor volumes; if statistically significant, then post-hoc testing was performed with correction for multiple comparisons. The log-rank test was used to compare the Kaplan–Meier curves of the progression-free survival endpoint of 300 mm^3^. One-way ANOVA with a post-hoc test was performed for secondary outcomes, such as changes in the tumor immunologic milieu and for in vitro experiments. Unpaired two tailed *t*-tests were performed to compare the PLA and free alendronate biodistribution and to compare the tumor weights after the efficacy endpoints. A *p*-value less than 0.05 was considered statistically significant. All statistical analyses were performed using GraphPad Prism software version 8 or higher.

## 3. Results

### 3.1. Characterization of Liposomal Formulations

The PLA and PLA-DiI formulations had comparable mean sizes ranging from 85–100 nm in diameter with the polydispersity index (PDI) between 0.05 and 0.08, the zeta potential between −16.3 and −33.0 mV, and pH between 6.8 and 7.2. The placebo liposomes had comparable characteristics except for larger PDI of 0.28. The final concentrations of phospholipids for the PLA and PLA-DiI were 44.0–48.8 mM and 26.4 mM, respectively, and the final concentrations of alendronate were 4.30–4.65 and 1.83 mg/mL, respectively (see Appendix A). The phospholipid concentration for the placebo liposome formulation was 25.6 mM.

### 3.2. In Vitro Cytotoxicity

In vitro MTT assays were performed to determine the direct cytotoxic activity of the PLA compared to free (non-liposomal) alendronate and the cytotoxic chemotherapy doxorubicin (positive control). The PLA had minimal cytotoxicity for B16-OVA melanoma cells, while free alendronate had moderate cytotoxicity (IC_50_~500 nM) compared to doxorubicin (IC_50_ ≤ 100 nM) (Figure 1A). Similar results were observed for the macrophages (Figure 1B), although the macrophages were less sensitive to doxorubicin than the B16-OVA cells. As an active immunotherapy comparator, we also tested anti-PD1, which also did not demonstrate any significant direct cytotoxicity in these cells (Appendix A). These results indicate that encapsulating alendronate in liposomes decreased its direct cytotoxicity and suggests that if PLA has anticancer activity, it is mediated by indirect effects on tumor cells, most probably through immune modulation.

### 3.3. Modulation of Immunological Pathways in Macrophages

Since the majority of systemically administered liposomes are internalized by macrophages in the liver and spleen [40], we sought to determine whether PLA had direct effects on macrophages, and whether their polarization states (M0, M1, or M2) had an impact on their responses to the PLA. Using murine bone marrow-derived macrophages, we evaluated the polarization state of macrophages by the mRNA expressions of iNOS and Arg-1, markers of M1 and M2 polarization, respectively. We found that when compared to vehicle (5% dextrose)-treated M0 macrophages, PLA significantly increased the expression of iNOS in M0 and M2 macrophages, while M1 macrophages showed no changes (Figure 2A). PLA also decreased the expression of Arg1 in M0 macrophages (Figure 2B). Further assessment of the cytokine markers showed that PLA significantly increased the expressions of IL-6, CXCL-10, and IL-10 in M0 and M2 macrophages without affecting the cytokine production in M1 macrophages (Figure 2C–E). The upregulation of IL-6 and CXCL-10 in macrophages at the protein level was also confirmed by ELISA (Figure 3A,B). Together, these results indicate that PLA induced M1-functionality even in M2-polarized macrophages.

### 3.4. PLA Is Stable In Vivo and Accumulates in Tumors and Tumor-Draining Lymph Nodes

Since M2-TAMs are major mediators of immunosuppression in the tumor microenvironment, the above effects of PLA on polarization indicate that PLA can promote anti-tumoral immune responses, especially if there is significant PLA accumulation in tumors. Previously, PLA labeled with radionuclides was shown to accumulate in tumors in animal models [41]. To verify this and further characterize biodistribution, free alendronate and PLA at a dose of 4 mg/kg of alendronate were administered intravenously to tumor-bearing mice in a B16-OVA melanoma model. Using LC-MS/MS, we found 6.5- to 161.8-fold higher alendronate concentrations in the tumor, spleen, liver, kidney, lung, and serum at 24 h post-administration of the PLA compared to the free alendronate (Figure 4A). Since free alendronate has a very short half-life and PLA has a long circulating time, we also compared the alendronate concentrations at 1 h and 48 h post-dose, respectively, which represents the approximate tissue T_max_ for each. We observed higher alendronate concentrations for the PLA compared to the free alendronate at these timepoints as well (Figure 4B). We next confirmed these findings using fluorescently labeled PLA (DiI-PLA) [16]. We detected DiI-PLA in the tumor, liver, and spleen (Figure 4C–E). Importantly, we observed DiI-PLA accumulation in tumor-draining inguinal lymph nodes, but not in non-tumor-draining axillary lymph nodes (Figure 4C–E). Together, these results indicate that the PLA was stable in vivo and enabled alendronate drug delivery to the tumor and tumor-draining lymph node tissues, the main sites of tumor-immune cell interactions, which are relevant for immunotherapy.

### 3.5. PLA Inhibits Tumor Growth

The efficacy of PLA was compared to a PD-1 immune checkpoint inhibitor antibody (anti-PD1), representing a clinical standard in cancer immunotherapy. In the B16-OVA melanoma model, anti-PD1 therapy did not significantly inhibit tumor growth, although there was a moderate improvement in progression-free survival (*p* = 0.06). On the contrary, PLA significantly inhibited tumor growth with mean endpoint tumor volumes of 111 mm^3^ for PLA compared to 371 mm^3^ for the control group (*p* = 0.0211) and prolonged progression-free survival (*p* = 0.0033) (Figure 5A,B). Tumor weights also showed a similar trend with the mean weight (249.3 mg) for the control group being significantly more (*p* = 0.0159) than the mean weight for the PLA (83.71 mg) (Figure 5C). These studies were terminated at a relatively early endpoint for humane reasons due to the development of tumor ulcerations and erythema that were likely related to the inflammatory activity of PLA.

### 3.6. PLA Remodels the Tumor Immunologic Milieu

Next, we inspected various immune cell populations in the tumor microenvironment of these two tumor models. Tumor-associated macrophages (TAMs) and myeloid-derived suppressor cells (MDSCs) are key mediators of immunosuppression in the tumor microenvironment (TME) and potent inhibitors of T-cell responses, while MDSCs are also a source of TAMs. In this melanoma model, we found that the PLA significantly inhibited TAM infiltration and polarization (by increasing the proportion of non-polarized M0-TAM in the TME) (Figure 6A,B), although it increased the total number of MDSCs and neutrophilic MDSCs (Figure 6C,D). We also noticed that the PLA significantly reduced the number of CD4^+^ helper T cells, which are immunosuppressive cell types, with no significant difference in the number of CD8^+^ cytotoxic T cells, which are the primary effector cell types against tumors (Figure 6E,F). Overall, these changes indicate a less suppressive immunologic milieu in response to PLA administration.

### 3.7. PLA Is Tolerable in Mouse Models

The main safety endpoints were treatment-related mortality and weight loss. In the B16-OVA tumor model, monotherapy with anti-PD1 or PLA at a dose of 4 mg or less was well tolerated, although a higher dose of PLA (5 mg/kg) was associated with more weight loss (Appendix A). No mortality was associated with PLA, while anti-PD1 was associated with a 5% mortality rate in the B16-OVA tumor model. The histopathology of the major organs performed on a subset of animals from each treatment group revealed no pathological changes in the heart and kidney and no significant differences in the immune cell infiltration in the lung, spleen, stomach, colon, or jejunum (Appendix A). PLA showed a tendency to increase inflammation in the jejunum, which was also grossly observable in post-mortem necropsies in the B16-OVA tumor model. In anti-PD1-treated mice, the liver showed mild inflammation (Appendix A).

## 4. Discussion

We showed herein that PLA induced macrophages to polarize towards an M1-functionality, whether they were M0- or M2-polarized macrophages. Free alendronate had negligible effects, indicating that liposome encapsulation is necessary for the observed immune modulation. PLA also remodeled the tumor microenvironment, and although the results are not identical to the in vitro studies, they similarly show a less immunosuppressive potential, as indicated by a decrease in TAM and helper T-cell infiltration [42,43,44]. On the other hand, we also observed that PLA increased MDSCs in tumors, which could be due to the inhibition of their differentiation into immunosuppressive TAMs and other cells. Although MDSCs may have a role in promoting tumor progression and immunosuppression, we still observed tumor growth inhibition with PLA. It is possible that PLA also alters the functionality of these MDSCs in the same way as it affects the functionality of macrophages, thus inhibiting their protumoral phenotypes. It has been reported that small molecule inhibitors of PI3Kγ (such as IPI-145) can suppress MDSCs’ pro-tumoral activities, but whether PLA acted similarly to IPI-145 warrants further investigation [45,46,47,48].

Another important new finding from our study is the preferential accumulation of PLA in tumor-draining lymph nodes as compared to unaffected lymph nodes. The tumor-draining lymph node, along with the local tumor, are the primary sites to induce antitumor T-cell response as well as the primary sites of tumor-induced immunosuppression. The negligible inhibitory effect in vitro on tumor cell growth coupled with the immune responses induced by PLA, indicates that PLA acts primarily as an immunotherapeutic agent by modulating macrophage functionality rather than through direct antiproliferative effects on tumor cells. Our finding that PLA enhanced alendronate drug delivery to tumors in the melanoma and HPV-induced TC-1 tumor models is consistent with other reports on murine models of mammary carcinoma [41]. The accumulation of PLA in tumors and tumor-draining lymph nodes further suggests that PLA could enhance the efficacy of other immunotherapies.

We also observed a decrease in CD4^+^ helper T cells but not in cytotoxic T cells in tumors treated by PLA. We believe that these are indirect effects of PLA. The repolarization of TAMs by PLA toward an M1-like phenotype would lessen immunosuppression orchestrated by M2-like macrophages in the tumor microenvironment. M2-polarized macrophages sustain the proliferation of regulatory T cells [49], an immunosuppressive subtype of CD4^+^ T cells, which could explain why the PLA repolarization of macrophages towards an M1 phenotype would decrease helper T-cell populations without affecting cytotoxic T cells. Macrophage M2 polarization is also associated with metabolic reprogramming characterized by increased oxidative phosphorylation, among other changes, which perpetuates hypoxia and suppresses antitumoral T-cell responses in the tumor microenvironment [50]. Metabolic reprogramming of the tumor microenvironment using nanoparticles has been reported to promote T-cell tumoricidal activity and decrease the expression of immuno-inhibitory molecules, such as PD-L1 [51], supporting this as a potential mechanism of PLA immune modulation.

PLA has additional mechanisms of immune stimulation in humans and primates that are not readily assessed in the current murine models. The inhibition of farnesyl pyrophosphate synthase (FPPS) by alendronate results in intracellular accumulation of isopentenyl pyrophosphate (IPP), a mevalonate metabolite that can covalently bind to adenosine monophosphate (AMP). This forms an analog that is recognized as a phosphoantigen by gamma-delta (γδ) T cells, specialized human T lymphocytes that include subtypes such as Vγ9Vδ2 T cells [52]. Phosphoantigens play an important role in the recognition and destruction of cancer cells by γδ T cells, resulting in their expansion and activation in vitro [53,54] and in vivo [55]. In humans, amino-bisphosphonates, such as alendronate, directly stimulate cellular expansion and interferon-gamma (IFNγ) secretion in γδ T cells obtained from the peripheral blood of healthy volunteers, whereas the non-amino-bisphosphonates, clodronate and etidronate, are inactive [56]. Patients treated with pamidronate, an amino-bisphosphonate for bone resorption, have been reported to experienced flu-like acute phase reactions along with an increase in circulating γδ T cells [57]. Another amino-bisphosphonate, zoledronate, also generates an acute phase immune response in patients [58,59], and is a potent stimulator of proliferation and activator of γδ T cells [60]. PLA has been reported to increase the therapeutic efficacy of γδ T cells in preclinical models [18,19,61]. Pre-treatment with PLA, but not placebo liposomes or free alendronate, significantly increased the trafficking of γδ T cells to mammary tumors [62]. The anticancer efficacy of the combination of PLA and adoptive γδ T-cell therapy was investigated in human xenograft models of ovarian cancer (IGROV-1 and SKOV-3) [19]. PLA was found to be highly effective as a sensitizing agent, enhancing γδ T-cell-mediated regression of established murine ovarian tumors compared to adoptive cell therapy without PLA.

The B16-OVA melanoma model expresses ovalbumin as a tumor neoantigen and is considered an immunogenic tumor model. Nonetheless, we observed that PD-1 blockade alone had only modest efficacy when treatment was started after the tumors had time to become established (e.g., tumor volume ≥ 50 mm^3^), likely due to the inhibitory effects of TAMs on effector T cells. In contrast, PLA had significant tumor inhibition effects. This suggests that targeting TAM functionality with PLA has the potential to be efficacious in patients with tumors that are resistant to immune checkpoint inhibitor therapies. In the future, we plan to perform studies to test the efficacy of PLA in combination with anti-PD1 and other immunotherapies in tumor-bearing mice, but additional work is needed to identify the optimal dose, sequence, and dosing interval for combination immunotherapy, as these could be critical factors determining whether the effects of the combined therapies will be additive, synergistic, or even antagonistic.

Overall, PLA and anti-PD1 were both well-tolerated. Mild hepatic inflammation was seen by histopathology in anti-PD1 but not in the other treatments; this is in agreement with what patients experience when receiving immune checkpoint inhibitor therapies [63]. Although colitis is a frequent immune-related adverse event in patients [63], the same was not visualized in our studies. On the contrary, the PLA and anti-PD1 treatments seemed to affect the jejunum more extensively in mice. Indeed, the results of anti-PD1 therapies and colitis models in mice have been controversial [64], suggesting that murine models do not recapitulate all immune-related adverse events encountered in humans.

## 5. Conclusions

Our study has demonstrated that PLA is an efficacious immunotherapy in a murine model of aggressive melanoma with an established tumor microenvironment. PLA has the potential to induce M1-polarization in macrophages and to alleviate tumor-associated immunosuppression. Furthermore, it accumulates in tumors and tumor-draining lymph nodes, both of which are critical target sites for stimulating potent antitumor immunity. The beneficial effects of PLA on macrophage and T-cell functionality suggest potential synergy with other immunotherapies. Based on the animal data and clinical studies with various alendronate formulations, treatment with PLA is likely to be rapidly translatable, although a safe and effective dose window in humans is difficult to predict due to the interspecies differences in the natural antitumor immunity response. We believe that PLA warrants further clinical development, and these clinical trials should incorporate tumor and blood biomarkers and, when possible, also include immunophenotyping studies to verify the anti-immunosuppressive effects of PLA in humans. 

## Figures and Tables

**Figure 1 biomolecules-13-01309-f001:**
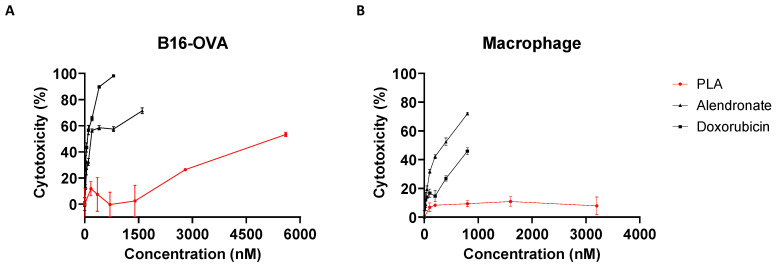
In vitro drug cytotoxicity. (**A**) B16-OVA melanoma cells, (**B**) bone marrow-derived macrophages were treated for 48 h with liposomal alendronate (PLA), alendronate, or doxorubicin (positive control) at various concentrations. Cytotoxicity was assessed using the MTT assay. Each treatment condition was performed in duplicate or triplicate; coefficients of variation were <10% for all replicates in all experiments. Data shown are mean ± SEM from one representative experiment; at least two experimental replicates were conducted.

**Figure 2 biomolecules-13-01309-f002:**
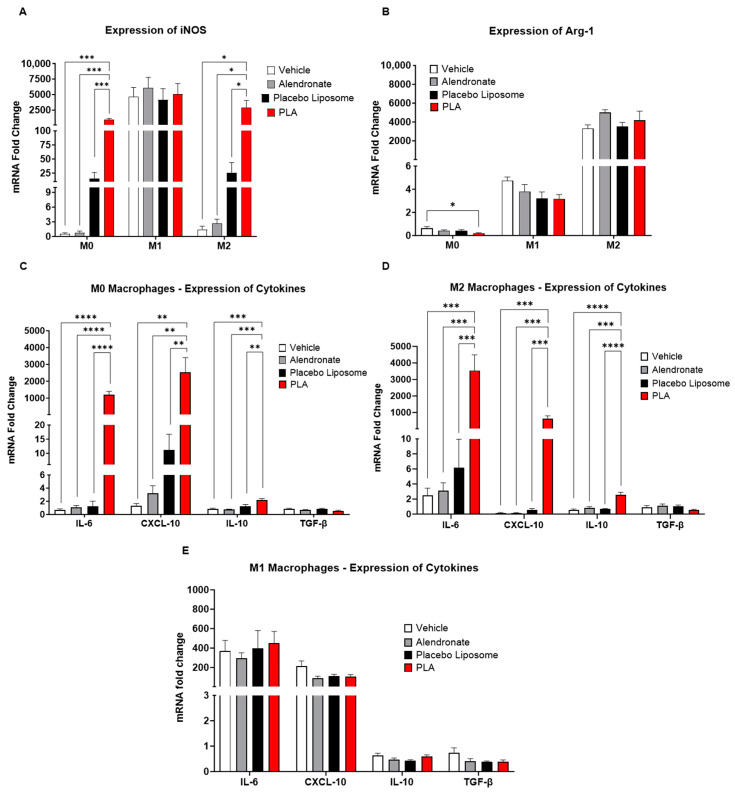
Liposomal alendronate induced M1 polarization and inflammatory markers in M0 and M2 bone marrow-derived macrophages (BMDMs). (**A**) Expression of iNOS as a marker of M1 polarization, (**B**) expression of Arg1 as a marker of M2 polarization, (**C**) expression of cytokines in unpolarized M0 macrophages, (**D**) expression of cytokines in M2-polarized macrophages, and (**E**) expression of cytokines in M1-polarized macrophages were determined by RT-qPCR. Fold changes were normalized to vehicle-treated unpolarized M0 cells. Results are expressed as mean ± SEM (*n* = 6/group from two experimental replicates); one-way ANOVA followed by Tukey’s multiple comparisons analyses; all pair-wise comparisons were performed, and only significant results are shown (* *p* < 0.05, ** *p* < 0.01, *** *p* < 0.001, **** *p* < 0.0001). PLA = liposomal alendronate.

**Figure 3 biomolecules-13-01309-f003:**
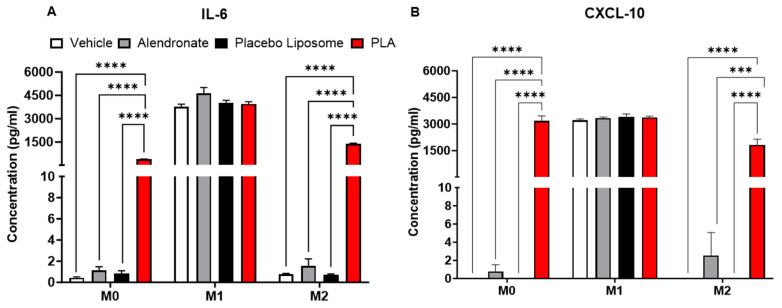
Levels of (**A**) IL-6 and (**B**) CXCL-10 production in unpolarized M0, M1-polarized, and M2-polarized bone marrow-derived macrophages (BMDMs) were confirmed by ELISA analyses of the cell culture supernatants from these experiments. Results are expressed as mean ± SEM (*n* = 6/group from two experimental replicates); one-way ANOVA followed by Tukey’s multiple comparisons analyses; all pair-wise comparisons were performed and only significant results are shown (*** *p* < 0.001, **** *p* < 0.0001). PLA = liposomal alendronate.

**Figure 4 biomolecules-13-01309-f004:**
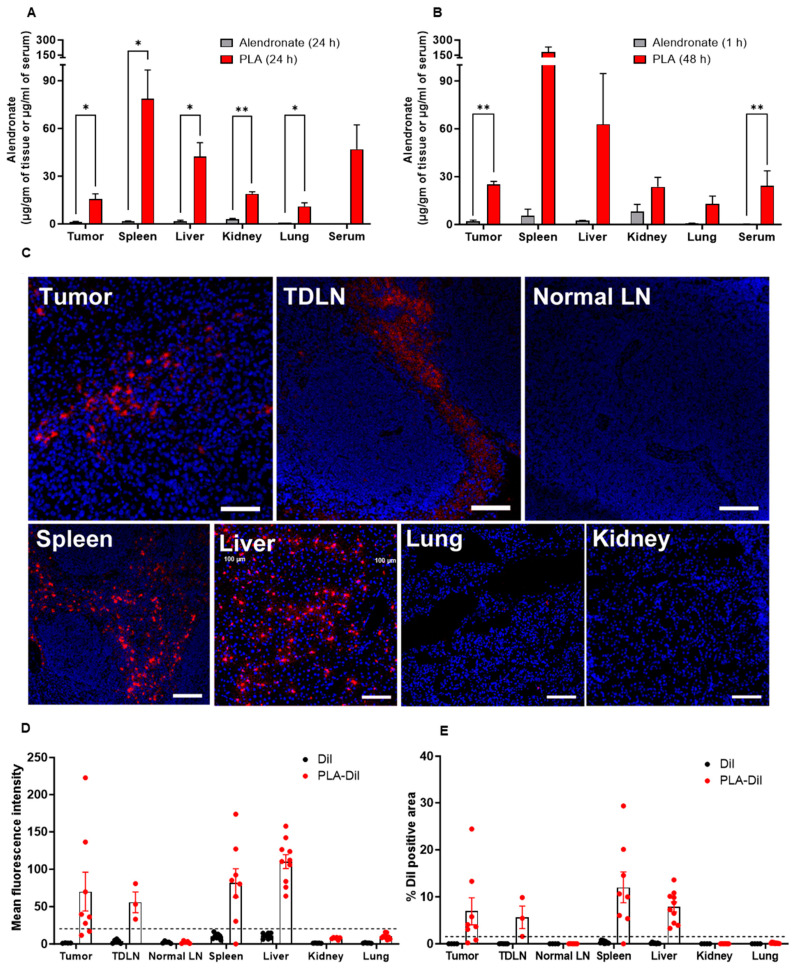
Biodistribution of liposomal alendronate (PLA) in tumor-bearing mice. LC-MS/MS quantitation of alendronate concentrations in different tissues and serum at (**A**) 24 h for both treatments, and (**B**) at 1 h (free alendronate) or 48 h (PLA) after intravenous dosing (*n* = 2–4/per group). Data represent mean ± SEM; unpaired two tailed *t*-tests; **p* < 0.05, ***p* < 0.01. (**C**) Representative fluorescence microscopy images of DiI-labeled PLA in different tissues at 24 h post-dose. (**D**) Mean DiI fluorescence intensity and (**E**) quantification of the proportion of tissue area that was DiI-positive for PLA-Dil and Dil only groups; each point represents one image. TLDN, tumor-draining lymph node; LN, lymph node.

**Figure 5 biomolecules-13-01309-f005:**
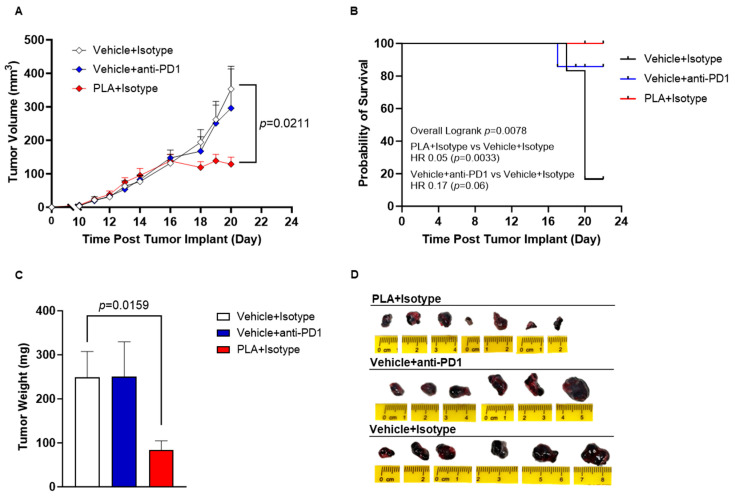
Antitumor efficacy of liposomal alendronate (PLA) in B16-OVA melanoma. Mice were implanted with B16-OVA cells, and then randomized (*n* = 6–7 mice/group) to treatments that were initiated when tumors were ~50 mm^3^. Tumor volume (**A**), Kaplan–Meier curves (**B**) and tumor weight (**C**) data show that PLA significantly reduced tumor growth in B16-OVA melanoma model. Images of whole tumor at endpoint from each treatment group (**D**). Data represent mean ± SEM; Two-way ANOVA with Dunnett’s test for tumor volume analyses; Logrank test for the Kaplan–Meier curve; unpaired two tailed T-test for tumor weights.

**Figure 6 biomolecules-13-01309-f006:**
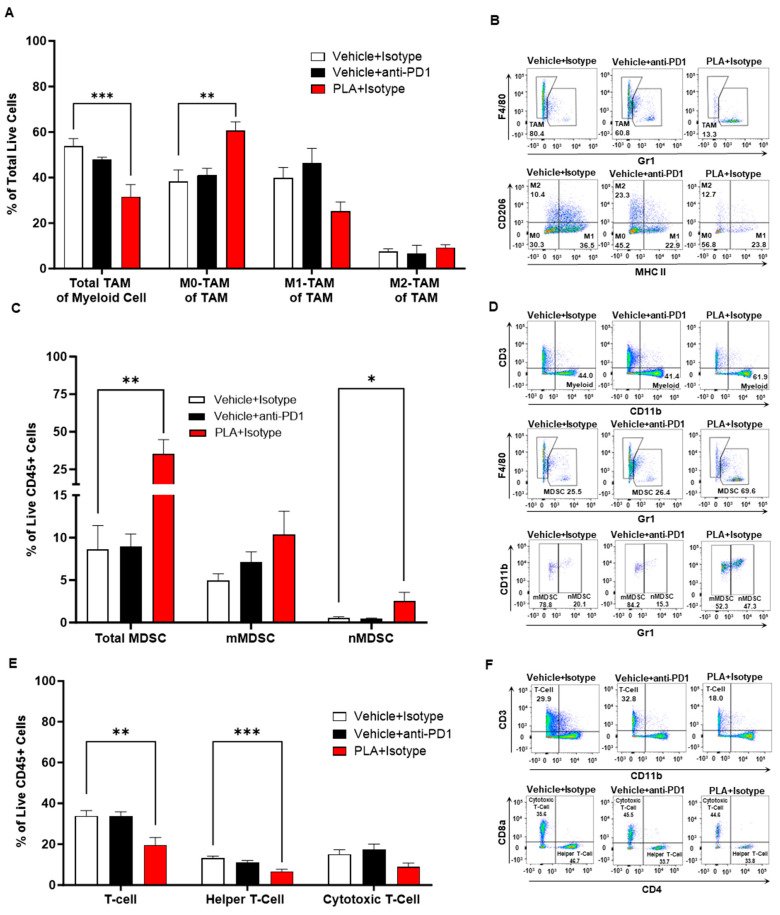
B16-OVA tumor immunological milieu. (**A**,**B**) In a B16-OVA melanoma tumor model, PLA treatment reduced the total tumor-associated macrophages (TAM) in the tumor microenvironment (TME). Analyses of the subpopulation of TAM revealed that PLA increased non-polarized TAM, but the effects on M1- or M2-polarized TAM were not significantly different than the vehicle group. (**C**,**D**) Treatment with PLA increased the total myeloid-derived suppressor cells (MDSC) and the subset of neutrophilic myeloid-derived suppressor cells (nMDSC) in the TME. (**E**,**F**) The total T-cell population was decreased with PLA treatment. Further analyses of the subpopulation revealed that the helper T cells were decreased without any effect on the cytotoxic T cells. *n* = 6 for PLA + isotype; *n* = 7 for vehicle + anti-PD1; *n* = 6 for vehicle + isotype. Mean ± SEM shown; one-way ANOVA with Dunnett’s test. * *p* < 0.05, ** *p* < 0.01, *** *p* < 0.001.

## Data Availability

Data is contained within the article and Appendix A.

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
