# Peer review of "Pegylated Liposomal Alendronate Biodistribution, Immune Modulation, and Tumor Growth Inhibition in a Murine Melanoma Model"

_biomolecules, 2023, doi:10.3390/biom13091309_

Round 1

Reviewer 1 Report

The authors compiled a comprehensive manuscript that details the cancer immuno-therapeutic potential of Alendronate using PEGylated liposome. The manuscript describes formulation preparation, in vitro and in vivo evaluation, and tumor study. Overall, the study is well-researched, and supported by compelling evidences, so I would recommend acceptance of the manuscript with some revisions needed.

1.       The author mentions that the PLA composition is similar to Doxil. However, Doxil was successful primarily because it lowers the doxorubicin concentration in the heart, reducing cardiotoxicity but not necessarily increasing the tumor-targeting effect of doxorubicin. It might be questionable on the efficiency of tumor targeting effect via EPR effect using “stealth liposome” in humans.

2.       Some liposome formulations have pretty high PDI (0.289) and varied sizes; it will be better to have the tabulated results of the size and PDI information for each formulation.

3.       Since this manuscript has focused on the biodistribution study, method validation of LCMS bioanalysis assays in different buffers/biological (e.g., chromatograms showing specificity, sensitivity, etc.) could be added to the supplemental information. Also, how did you know tissue Tmax for PLA is 48h? Is there any plasma or tissue PK of PLA done? It’s OK if not.

4.       For the in vitro cytotoxicity study, I would suggest the authors put in the actual curve of in vitro cytotoxicity study (dose-response curve) in addition to the viability ratio curve. Since IC50 of PLA increases in both B16 and BMDM, this suggests that there is limited Alendronate released or cellular uptake/internationalization of the PLA over the 24 h period. The cellular uptake study and/or kinetic release study of the PLA will be very useful in elucidating the behavior of PLA.

5.       How to maintain the PLA to have only the immune modulatory effect but not the ablation effect to the TAM in tumors? Is it dose dependent?

What is the potential reason to the decrease of helper T cells frequency but not the cytotoxic T cells in tumors treated by PLA?

All flow-related information, such as antibodies dilution, clone, gating strategies, etc. could go to supplemental information, which I didn't see in the manuscript, though. The PCR primer sequences table could also go to supplemental information.

Author Response

Comments and suggestions for authors: The authors compiled a comprehensive manuscript that details the cancer immuno-therapeutic potential of Alendronate using PEGylated liposome. The manuscript describes formulation preparation, in vitro and in vivo evaluation, and tumor study. Overall, the study is well researched, and supported by compelling evidences, so I would recommend acceptance of the manuscript with some revisions needed.

  1. The author mentions that the PLA composition is similar to Doxil. However, Doxil was successful primarily because it lowers the doxorubicin concentration in the heart, reducing cardiotoxicity but not necessarily increasing the tumor-targeting effect of doxorubicin. It might be questionable on the efficiency of tumor targeting effect via EPR effect using “stealth liposome” in humans.

Response: We appreciate the reviewer’s comment and the reviewer is correct that Doxil is well known to reduce cardiotoxicity of doxorubicin. However, another important property of liposomes with Doxil-like PEGylated composition (known also as Stealth) is their passive tumor targeting properties via EPR effect (Golombek et al., 2018: https://pubmed.ncbi.nlm.nih.gov/30009886/; Gabizon et al., 2020: https://pubmed.ncbi.nlm.nih.gov/32526450/). Passive tumor targeting of these long-circulating liposomes has been documented also in humans (Harrington et al., 2001:https://pubmed.ncbi.nlm.nih.gov/11234875/) although its extent is highly variable (Lee et al., 2017: https://pubmed.ncbi.nlm.nih.gov/28298546/). Based on the preclinical and clinical evidence of EPR effect for this stealth liposome, we chose a Doxil-like formulation to deliver alendronate since our aim is to enhance alendronate delivery to tumors. We have added this to the Introduction to better support our rationale for using this liposomal carrier (Page 2, Lines 98-100).

  1. Some liposome formulations have pretty high PDI (0.289) and varied sizes; it will be better to have the tabulated results of the size and PDI information for each formulation.

Response: We apologize for the confusion. All PLA formulations were comparable with low PDI, the high PDI was observed in the placebo liposomes only. We have clarified these results (Page 7, Lines 301-306) and added to the supplemental materials a table of the characterization data for each PLA batch used in this study (Supplemental Table S2).

  1. Since this manuscript has focused on the biodistribution study, method validation of LCMS bioanalysis assays in different buffers/biological (e.g., chromatograms showing specificity, sensitivity, etc.) could be added to the supplemental information. Also, how did you know tissue Tmax for PLA is 48h? Is there any plasma or tissue PK of PLA done? It’s OK if not.

Response:

The developed LC-MS/MS method was based on a published method (reference 37, doi: 10.3389/fimmu.2022.840029), which we further assessed for suitability through evaluation of linearity and specificity within matrix. Additionally, calibration curve and quality control samples were analyzed with each run for accuracy throughout the analysis.  The concentration of each of the calibration curve standard, when back calculated, was determined to be within  ±15% from its nominal value except at LLOQ, where it was determined to be within ± 20%. The back-calculated concentrations for the quality control samples were determined to be within ± 15% of their nominal value. We have added these additional details to the manuscript methods (Page 5-6, Lines 253-259) and we have added representative chromatograms to the supplemental materials (Supplemental Figure S1-S2).

We had previously performed a study with radiolabeled alendronate (La-Beck et al., 2021: https://pubmed.ncbi.nlm.nih.gov/31874280/) that showed a strong correlation between clearance of pegylated liposomal alendronate and that of pegylated liposomal doxorubicin, which both use the same liposomal carrier. Since the optimal Tmax for the latter (Doxil) is between 48 and 72 h (Gabizon et al., 2003: https://pubmed.ncbi.nlm.nih.gov/12739982/), we chose this time point also to measure alendronate levels in tissues.

  1. For the in vitro cytotoxicity study, I would suggest the authors put in the actual curve of in vitro cytotoxicity study (dose-response curve) in addition to the viability ratio curve. Since IC50 of PLA increases in both B16 and BMDM, this suggests that there is limited Alendronate released or cellular uptake/internationalization of the PLA over the 24 h period. The cellular uptake study and/or kinetic release study of the PLA will be very useful in elucidating the behavior of PLA.

Response: We appreciate the reviewer’s suggestion and have replaced the viability ratio curve with the cytotoxicity curve in the manuscript (Figure 1). Based on our prior studies of PLA, there is negligible release of alendronate. We agree that cellular uptake studies can be useful to further understand the behavior of PLA and we will do this in our future work with PLA.

  1. How to maintain the PLA to have only the immune modulatory effect but not the ablation effect to the TAM in tumors? Is it dose dependent?

What is the potential reason to the decrease of helper T cells frequency but not the cytotoxic T cells in tumors treated by PLA?

All flow-related information, such as antibodies dilution, clone, gating strategies, etc. could go to supplemental information, which I didn't see in the manuscript, though. The PCR primer sequences table could also go to supplemental information.

Response:

Yes, the reviewer is correct that the effects of PLA are dose dependent. In our prior studies, we observed that PLA at doses higher than that used in this study was associated with significant systemic morbidity and mortality accompanied by signs suggestive of cytokine storm, while PLA at 4 mg/kg or lower was well-tolerated and maintained immune stimulatory effects. Unlike clodronate, which is a toxic ATP analogue, alendronate inhibits the mevalonate pathway. We believe that this is the reason why we did not observe macrophage ablation with PLA while liposomal clodronate readily depletes systemic macrophages.

We believe that the effects on T cells in the tumors are indirect effects of PLA. Repolarization of TAM by PLA toward an M1-like phenotype would lessen immunosuppression orchestrated by M2-like macrophages in the tumor microenvironment. M2-polarized macrophages sustain proliferation of regulatory T cells (doi: 10.3389/fimmu.2021.710406) which could explain why PLA repolarization of macrophages towards an M1-phenotype would decrease CD4+ T cell populations without affecting CD8+ cytotoxic T cells. Macrophage M2-polarization is also associated with metabolic reprogramming characterized by increased oxidative phosphorylation, among other changes, which perpetuates hypoxia and suppresses antitumoral T cell responses in the tumor microenvironment (doi: 10.3389%2Ffimmu.2022.840029). Metabolic reprogramming of the tumor microenvironment using nanoparticles has been reported to promote T cell tumoricidal activity and decrease expression of immuno-inhibitory molecules such as PD-L1 (doi: 10.1016/j.jconrel.2022.11.004) supporting this as a potential mechanisms of PLA immune modulation. We have added this to our Discussion of PLA mechanisms of immune modulation (Page 13, Lines 465-478). 

We thank the reviewer for the helpful suggestion to move some of the methodology details to supplemental materials and we have done so. We apologize that the supplemental files were not available to the reviewer, we uploaded the file but it may not have been attached by the submission system. We have strived to correct this with the resubmitted files.

Reviewer 2 Report

In this research, the authors evaluated the possibility of Pegylated Liposomal Alendronate Biodistribution, Immunemodulation, and Tumor Growth Inhibition in a Murine Melanoma Model. Generally, it’s meaningful and interesting. In my opinion, the current version of this manuscript fits the scope of Biomolecules and could be accepted after major revision.

My specific comments are in detail listed below:

1.      In the introduction, the merits of using liposome as drug carrier should be more clearly discussed. Some references could be added including 10.1002/advs.202207608.

2.      All the font size of the Figures is small. The authors should revise or modify it.

3.      In the discussion, a better conclusion that conclude or predict how pegylated liposomal alendronate affect the immune status like T cells should be more clearly discussed. Some references could be added including 10.1016/j.jconrel.2022.11.004.

4.  Some minor mistakes existed in the references. Besides, some references are out of date. The authors should carefully check it.

5.  Some minor English usage errors exist. The authors should polish it.

6.  In Fig. 5D, it’s confusing why the image of tumors was separated rather than collected together?

Author Response

Reviewer 2

Comments and suggestions for authors: In this research, the authors evaluated the possibility of Pegylated Liposomal Alendronate Biodistribution, Immune modulation, and Tumor Growth Inhibition in a Murine Melanoma Model. Generally, it’s meaningful and interesting. In my opinion, the current version of this manuscript fits the scope of Biomolecules and could be accepted after major revision.

My specific comments are in detail listed below:

  1. In the introduction, the merits of using liposome as drug carrier should be more clearly discussed. Some references could be added including 10.1002/advs.202207608.

Response: We thank the reviewer for this helpful suggestion and have added a discussion of this reference (reference 21) to the Introduction to better support the merits of liposomes as drug carriers (Page 2, Lines 87-90).

  1. All the font size of the Figures is small. The authors should revise or modify it.

Response: The font size in the figures has been increased by 125%

  1. In the discussion, a better conclusion that conclude or predict how pegylated liposomal alendronate affect the immune status like T cells should be more clearly discussed. Some references could be added including 10.1016/j.jconrel.2022.11.004.

Response: We thank the reviewer for this helpful suggestion and have expanded on our discussion of how PLA could modulate functionality of T cells (Page 13, Lines 465-478) and have added the suggested reference (reference 51). 

  1. Some minor mistakes existed in the references. Besides, some references are out of date. The authors should carefully check it.

Response: We have corrected the errors in the references, and have replaced out of date references with updated ones.

  1. Some minor English usage errors exist. The authors should polish it.

Response: We have corrected the English usage errors.

  1. In Fig. 5D, it’s confusing why the image of tumors was separated rather than collected together?

Response: We apologize for the confusion. At study endpoint, we prioritized rapid processing of the tumor tissues in order to maximize viability of tumor-derived single cell suspensions for flow cytometric analyses. We also needed to rapidly collect and process all major organs for assessment of pathological inflammation. Because of this, the tumor images were taken as we collected them rather than after all animals were sacrificed and processed.

Reviewer 3 Report

 The manuscript presented for publication at Biomolecules by Islam et al. is reporting a delivery platform based on PEGylated HSPC/DSPE liposome for alendronate, which otherwise has low bio-availability in its free form, as a broad-spectrum anti-tumor immunotherapeutic agent. The reported long-circulating liposomal alendronate showed low in vitro cytotoxicity and induce M1 functionality even in M2 polarized macrophages. After injected intravenously, PLA are found to accumulate in tumor, TDLN, spleen and liver, but not in non-draining LN. More specifically, PLA are shown to supressed tumor growth in murine melanoma B16-OVA model and prolonged survival when compared to immune checkpoint blockade aPD1. Interestingly, PLA treatment in B16-OVA model increase total percentage of MDSC and mMDSC. Overall, the manuscript is well written and therefore I would suggest accepting the manuscript upon revision of the minor comments indicated below:

Minor comments:

1.     Not sure whether is system error related, I am not able to find the supplemental tables and figures. Please provide all relating supplementary information.

2.     The choice of experimental model: I could appreciate authors chose B16-OVA as it’s considered as an immunogenic tumor model. On the other hand, being an aggressive tumor B16-OVA often leaves a relatively short time frame to evaluate the therapeutics efficacy. In this manuscript the efficacy experiment in Fig.5 were terminated early timepoint due to tumor ulcerations, soon after the second dose of PLA or the third dose of aPD-1. Is there any explanation for higher tumor cell number was injected to mice for the efficacy experiment (106 cells vs. 0.5x106 cells in biodistribution experiment)?

3.     Would you expect the PLA behave similarly in the non-immunogenic parental B16-F10 tumor?

4.     Figure 6: Y-axis labels for A, C and E are very confusing and misleading. Please specify the population of cells referring to. For instance: A. Y axis only labelled with % of cells. But which population of cells that referred to? Total live cells, total live CD45+ cells? Also in E, are all three groups (T-cell, Helper T-cells and Cytotoxic T-cells) presented were calculated by % of CD45+? If that’s the case, since there was decrease of T-cell in PLA+Isotype treated group, it will not surprise that there’s also decrease in Th and CTL  found in the same group.

Author Response

Comments and suggestions for authors: The manuscript presented for publication at Biomolecules by Islam et al. is reporting a delivery platform based on PEGylated HSPC/DSPE liposome for alendronate, which otherwise has low bio-availability in its free form, as a broad-spectrum anti-tumor immunotherapeutic agent. The reported long-circulating liposomal alendronate showed low in vitro cytotoxicity and induce M1 functionality even in M2 polarized macrophages. After injected intravenously, PLA are found to accumulate in tumor, TDLN, spleen and liver, but not in nondraining LN. More specifically, PLA are shown to supressed tumor growth in murine melanoma B16-OVA model and prolonged survival when compared to immune checkpoint blockade aPD1. Interestingly, PLA treatment in B16-OVA model increase total percentage of MDSC and mMDSC. Overall, the manuscript is well written and therefore I would suggest accepting the manuscript upon revision of the minor comments indicated below:

Minor comments:

  1. Not sure whether is system error related, I am not able to find the supplemental tables and figures. Please provide all relating supplementary information.

Response:

We believe there was an error in the submission system that did not attach the supplemental file that we submitted. We have strived to correct this with the resubmitted files.

  1. The choice of experimental model: I could appreciate authors chose B16-OVA as it’s considered as an immunogenic tumor model. On the other hand, being an aggressive tumor B16-OVA often leaves a relatively short time frame to evaluate the therapeutics efficacy. In this manuscript the efficacy experiment in Fig.5 were terminated early timepoint due to tumor ulcerations, soon after the second dose of PLA or the third dose of aPD-1. Is there any explanation for higher tumor cell number was injected to mice for the efficacy experiment (106 cells vs. 0.5x106 cells in biodistribution experiment)?

Response: In our biodistribution study, we observed that the time from tumor cell injection to when the tumors were palpable was longer than expected therefore, we increased the number of cells that we injected for the efficacy studies. Once tumors were established (i.e., palpable), we did not see a difference in the growth rate of the untreated control tumors. The only difference was a shorter time to establishment. Tumor ulcerations were more prevalent among the PLA treated tumors compared to the other groups suggesting that this is due to the immune stimulatory effects of PLA in the tumor microenvironment, rather than the number of tumor cells that we injected.

  1. Would you expect the PLA behave similarly in the nonimmunogenic parental B16-F10 tumor?

Response: Since PLA mainly affects macrophages, and macrophages can coordinate not just T cell immune responses but also immune responses in innate immune cells such as NK and NKT, it is possible that PLA will have similar antitumoral effects in the B16-F10 tumor model.

  1. Figure 6: Y-axis labels for A, C and E are very confusing and misleading. Please specify the population of cells referring to. For instance: A. Y axis only labelled with % of cells. But which population of cells that referred to? Total live cells, total live CD45+ cells? Also in E, are all three groups (T-cell, Helper T-cells and Cytotoxic T-cells) presented were calculated by % of CD45+? If that’s the case, since there was decrease of T-cell in PLA+Isotype treated group, it will not surprise that there’s also decrease in Th and CTL found in the same group.

Response: We apologize for the confusion and have made the recommended changes to Figure 6 Y-axis labels.

Round 2

Reviewer 1 Report

Thanks for addressing all my comments. The in vitro cytotoxicity figure might be clearer if you use log scale for the dose concentration. For the flow cytometry gating strategy, I would recommend adding a duplet exclusion gate (e.g., FSC-H vs. FSC-A) as well since tumor tissue cells/immune cells will be "sticky" and tend to form cell aggregate. 

Otherwise, the manuscript looks good! 

Reviewer 2 Report

The current version of this manuscript could be accepted.